# The Double-Edged Sword of Oleuropein in Ovarian Cancer Cells: From Antioxidant Functions to Cytotoxic Effects

**DOI:** 10.3390/ijms24010842

**Published:** 2023-01-03

**Authors:** Stefania Scicchitano, Eleonora Vecchio, Anna Martina Battaglia, Manuela Oliverio, Monica Nardi, Antonio Procopio, Francesco Costanzo, Flavia Biamonte, Maria Concetta Faniello

**Affiliations:** 1Research Center of Biochemistry and Advanced Molecular Biology, Department of Experimental and Clinical Medicine, “Magna Graecia” University of Catanzaro, 88100 Catanzaro, Italy; 2Interdepartmental Centre of Services, “Magna Graecia” University of Catanzaro, 88100 Catanzaro, Italy; 3Department of Health Science, “Magna Graecia” University of Catanzaro, 88100 Catanzaro, Italy

**Keywords:** oleuropein, ovarian cancer, ROS

## Abstract

Oleuropein plays a key role as a pro-oxidant as well as an antioxidant in cancer. In this study, the activity of oleuropein, in an in vitro model of ovarian (OCCs) and breast cancer cells (BCCs) was investigated. Cell viability and cell death were analyzed. Oxidative stress was measured by CM-H2DCFDA flow cytometry assay. Mitochondrial dysfunction was evaluated based on mitochondrial reactive oxygen species (ROS) and GPX4 protein levels. Further, the effects on iron metabolism were analyzed by measuring the intracellular labile iron pool (LIP). We confirmed that high doses of oleuropein show anti-proliferative and pro-apoptotic activity on HEY and MCF-7 cells. Moreover, our results indicate that low doses of oleuropein impair cell viability without affecting the mortality of cells, and also decrease the LIP and ROS levels, keeping them unchanged in MCF-7 cells. For the first time, our data show that low doses of oleuropein reduce erastin-mediated cell death. Interestingly, oleuropein decreases the levels of intracellular ROS and LIP in OCCs treated with erastin. Noteworthily, we observed an increased amount of ROS scavenging enzyme GPX4 together with a consistent reduction in mitochondrial ROS, confirming a reduction in oxidative stress in this model.

## 1. Introduction

In recent years, a lot of evidence has been accumulated on the importance of a correct lifestyle in the field of human health and disease prevention. A central element of a healthy lifestyle is represented by the type of diet, and in this context, the importance of the Mediterranean diet has been repeatedly underlined, with the identification of the beneficial effects of many natural compounds. Among these, particular attention has been reserved to secoiridoids and their derivatives [1] of which oleuropein (OLE), contained in olive fruit, oil, and leaf, is the precursor [2].

OLE is the ester between 2-(3′,4′-dihydroxyphenyl) ethanol (hydroxytyrosol) and elenolic acid, a secoiridoid glycoside belonging to the class of coumarin components [3]. This secoiridoid is the main bioactive component present in the olive tree which confers the main bitter taste and resistance to the development of oil rancidity [4]. To date, a large number of scientific evidence suggests that OLE and its derivatives exert various protective effects against multiple pathologies such as neuro and cardiovascular diseases, diabetes mellitus, cancer, and chronic kidney disease, probably through a currently not fully elucidated antioxidant and anti-inflammatory activity [2,5,6,7,8,9,10,11]. In addition, OLE presents other peculiar features such as autophagy induction, amyloid fibril growth inhibitor, and, finally, anticancer activity [7].

Indeed, OLE, as a natural anticancer and pro-oxidant agent, acts as an inhibitor of cell growth and inducer of apoptosis in different cancer models, including breast cancer [12,13,14], melanoma [6], colon carcinoma [15], and hepatocellular carcinoma [16]. As a pro-oxidant compound, OLE promotes cell damage by increasing the amounts of reactive oxygen species (ROS) thus contributing to cell death [12,17]. Among other models, in ovarian cancer, the role of OLE is poorly understood. It was shown that OLE increases the sensitivity to radiation of human ovarian cancer cell lines [18] and regulates miRNA expression in ovarian cancer cells, reducing cisplatin resistance [19]. 

Recently, in ovarian and breast cancer, the pro-oxidant activity of OLE via mitochondrial impairment was shown [20].

In all these experimental models, OLE exerts its activity depending on the cell type, time, and concentration of exposure. OLE might display not only pro-oxidant but also antioxidant properties [21,22,23]. The double-edged sword of OLE is a typical property of most phytochemicals of the Mediterranean diet. Evidence support that these natural compounds act as hormesis, triggering a biphasic dose-response of biological systems [24]. It is demonstrated that low-dose exposures induce stimulatory or beneficial effects, while high-dose exposures lead to inhibitory or toxic effects.

The ability of OLE to chelate metal ions such as iron promotes its antioxidant activity [3]. In human peripheral leukocytes and HepG2 cells, OLE reduces H_2_O_2_-induced DNA damage suggesting its potential activity as a radical scavenger and metal ions chelator [25,26]. Previous findings demonstrated that OLE inhibits the formation of intracellular ROS by increasing the expression of intracellular antioxidant enzymes such as superoxide dismutase (SOD), glutathione peroxidase (GPx), glutathione reductase (GRx), and catalase (CAT). The OLE-induced activation of GPX appears mediated by its capacity to increase the intracellular amount of the reduced glutathione (GSH), which acts as a major co-factor of this antioxidant enzyme. The effects of OLE on CAT and SOD, instead, remain to be defined [27]. Further, in vitro studies indicate that OLE acts both as an antioxidant and pro-oxidant on non-tumorous as well as tumor prostate cells [28].

Despite a large body of studies, the molecular basis of how OLE affects ROS homeostasis in cancer cell models and the definition of the ROS-related pathways influenced by polyphenols such as oleuropein, are still not fully elucidated.

In this study, we analyzed the effects of high and low concentrations of OLE in in vitro models of ovarian (OC) and breast cancer (BC). We confirmed that high concentrations of OLE show anti-proliferative and pro-apoptotic activity in HEY and MCF-7 cells. Of note, OLE as a pro-oxidant agent causes the generation of free iron levels and ROS in HEY cell lines. Moreover, differently from previous investigations that showed OLE as a pro-oxidant agent in OC and BC conditions, we raised the hypothesis of a protective role exerted by OLE in OC cells but not in BC cells. Further, at lower doses, OLE acts as an antioxidant and iron chelator in oxidative stress conditions induced by erastin in OC cells, counteracting cytotoxic effects induced by erastin and reverting mitochondrial dysfunction. To the best of our knowledge, this is the first evidence of the action of oleuropein related to its antioxidant activity in ovarian cancer cells, specifically focalizing interest in the improvement of mitochondrial functionality.

## 2. Results

### 2.1. High Doses of Oleuropein Inhibit Cell Viability Inducing Cell Cycle Alteration and Apoptosis in HEY Cells

Previous findings suggest that oleuropein selectively reduces cell proliferation in different cancer types [19,22,29]. Here, to assess the effects of OLE on ovarian cancer cells, we treated HEY cells with growing concentrations of OLE (100 µM, 200 µM, and 400 µM) for 24 h. We demonstrated that, except for the lowest concentration (100 µM), OLE administration triggers cytotoxic effects, as demonstrated by the significant reduction in cell viability measured by both Trypan Blue dye exclusion (200 µM and 400 µM) (~59%, *p*-value < 0.0001; ~37%, *p*-value < 0.0001, respectively; Figure 1A) and Cell Titer-Glo Luminescent assays (200 µM and 400 µM) (RLU = 1570, *p*-value < 0.0001; RLU = 668.4, *p*-value < 0.0001, respectively; Figure 1B). Furthermore, flow cytometry analysis shows that, at the highest concentration (400 µM), OLE also exerts a cytotoxic effect, as suggested by the increase in the PI^+^ HEY cells (45.6%) (Figure 1C). Then, in order to better investigate the cytotoxic effect of OLE, we focused our attention on the concentration of 400 µM. As represented in Figure 1D, OLE induces a significant decrease in S-phase (from 36.63% to 12.97%; *p*-value = 0.03) in parallel with an increase in both the subG1 population (from 0% to 3.57%; *p*-value = 0.01) and the G2/M population (from 18.3% to 46.1%; *p*-value = 0.005). These data indicate that, when used at high concentrations, oleuropein leads to a slowing transition in the S-phase and a significant accumulation of cells in the subG1 and G2/M phases in HEY cells. Finally, by performing Annexin V binding assay, we observed that 400 µM oleuropein for 24 h increases the percentage of apoptotic cells (from 2.3% to 54.8%) of HEY-treated cells compared to control cells (Figure 1E). Altogether, these results strongly support the idea that high doses of oleuropein mediate HEY ovarian cancer cell growth inhibition by promoting cell cycle arrest and apoptosis.

### 2.2. Oleuropein Acts as Pro- or Antioxidant in a Dose-Dependent Manner in HEY Cells

In vitro and in vivo studies have demonstrated that oleuropein can exert both antioxidant and pro-oxidant activity [21,30,31,32]. OLE’s antioxidant properties stem from its capacity to chelate metal ions, such as iron. The resulting complexes may act by reducing the intracellular free labile iron pool (LIP) and its possible redox-related toxicity. Indeed, when present in excess within the cells, iron disrupts redox homeostasis inducing ROS formation [33]. Hence, to investigate whether OLE modulates the LIP content also in ovarian cancer, we treated HEY cells with both the lowest (100 µM) and the highest dose (400 µM) for 24 h. Flow cytometric analysis with Calcein-AM staining highlighted that treatment with the lowest dose of OLE reduces intracellular LIP levels (Figure 2A,B). In agreement, the same concentration of OLE determines a significant reduction in intracellular total ROS as shown by the H2DCFDA Cellular ROS Detection Assay (Figure 2C,D). Interestingly, when used at the highest dose, OLE shows an opposite effect, as demonstrated by the significant increase in LIP (Figure 2E,F) and ROS levels (Figure 2 G,H) in HEY cells compared to the control. Overall, these results suggest that, in our cell model of ovarian cancer, oleuropein acts as a pro-oxidant agent at high doses, while acting as an antioxidant at low doses.

### 2.3. Low Doses of Oleuropein Mitigate Intracellular Iron and ROS Accumulation and Counteract Cytotoxic Effects Induced by Erastin

To better investigate the role of oleuropein as an antioxidant and iron chelator molecule in ovarian cancer, we treated HEY cells with the ferroptosis inducer (FIN) erastin [34,35], in order to induce intracellular iron overload and, consequently, ROS production. 

Morphological observation under an inverted phase-contrast microscope (Figure 3A) and CellTiter-Glo Luminescent Cell Viability Assay (Figure 3B) indicate that erastin treatment (2.5 µM for 24 h) reduces the viability of HEY cells (from RLU = 5539 to RLU = 234.8; *p*-value < 0.0001; Figure 3B), while the co-treatment with OLE (100 µM for 24 h) significantly restores cell viability (from RLU = 655 to RLU = 234.8; *p*-value = 0.047; Figure 3B). In agreement, flow cytometric analyses show that OLE also prevents the effects of erastin on cell cycle distribution and the cell death of HEY cells. Indeed, erastin alone significantly raises the subG1 population (from 0.13% to 10%; *p*-value = 0.003) and decreases the S-phase (from 38.07% to 28.03%; *p*-value = 0.03), while the co-treatment of erastin–OLE restores cell cycle distribution by lowering the subG1 phase (from 10.5% to 4.15%; *p*-value = 0.008) and enhancing the S-phase (from 28.03% to 38.23%; *p*-value = 0.02; Figure 3C). Furthermore, the administration of erastin alone causes 35% of cell death, while the combined treatment erastin–OLE damps cell mortality rate to 26.9% (Figure 3D). The flow cytometric analysis of LIP and ROS content clearly shows that, as expected, erastin treatment is associated with the increase in LIP and, consequently, in total ROS (from 100% to 135.8%; *p*-value = 0.026; Figure 4A,B) (from 100% to 160%; *p*-value = 0.0003; Figure 4C,D) that are both significantly mitigated in HEY cells treated with the combination of erastin–OLE (from 135.8% to 104.3%; *p*-value = 0.04; Figure 4A,B) (from 160% to 140%; *p*-value = 0.04; Figure 4C,D). As previous findings highlight that erastin promotes the accumulation of mitochondrial-derived ROS (mitoROS) in HEY cells and, thus, is associated with mitochondrial dysfunction [36,37], we decided to specifically quantify mitoROS in both HEY cells treated with erastin alone or erastin–OLE. Flow cytometric analysis by using a MitoSOX Red probe shows a significant increase in fluorescence intensity in erastin-treated cells compared to untreated cells (from 100% to 182%; *p*-value = 0.002; Figure 4E,F); intriguingly, the combined treatment erastin–OLE halves fluorescence intensity compared to erastin treatment alone (from 182% to 124.8%; *p*-value = 0.028; Figure 4E,F), thus suggesting that OLE functions as an antioxidant also on ROS derived from mitochondria. This hypothesis is further suggested by the analysis of glutathione peroxidase 4 (GPX4) amounts, a well-known antioxidant enzyme playing a crucial role in protecting mitochondria from oxidative damage [38]. A representative Western blotting analysis demonstrated a significant decrease in GPX4 protein levels in erastin-treated cells compared to untreated cells; interestingly, the combined treatment of erastin–OLE restores GPX4 protein levels (Figure 4G). Overall, these data suggest that low doses of OLE counteract the cytotoxic effects triggered by erastin by mitigating ROS production caused by intracellular iron accumulation.

### 2.4. Oleuropein Uniquely Acts as a Pro-Oxidant at High Doses in MCF-7 Breast Cancer Cells

In order to assess whether the double function of OLE observed in ovarian cancer cells is a cell-specific feature or rather a more general phenomenon, we analyzed the effects of different concentrations of OLE on MCF-7 breast cancer cells in which the anticancer role of OLE has been already evaluated both in vitro and in vivo [29,39,40]. Similarly to HEY cells, MCF-7 cells treated with the lowest concentration of OLE (100 µM) result in only being slightly affected, while the % of viable cells is significantly reduced upon treatment with 200 µM–400 µM (Figure 5A,B). Flow cytometric analysis shows a significant increase in cell death in MCF-7 cells treated with OLE, particularly at the concentrations of 200 µM (35.8% PI^+^ cells) and 400 µM (57.9% PI^+^ cells) for 24 h (Figure 5C). Then, once again, we focused our attention on the highest concentration. As represented in Figure 5D,E, 400 µM OLE induces the increase in the subG1 population from 0.4% to 5.58% (*p*-value = 0.02), and the increase in the apoptotic rate from 9.24% to 23.5% of MCF-7 treated cells compared to untreated cells. Once the pro-oxidant role of OLE used at the highest concentration had been established, we proceeded to evaluate the possible antioxidant role of OLE in the MCF-7 context. Flow cytometric analyses showed that when used at the lowest dose (100 µM) OLE treatment is not associated with LIP chelation (Figure 5F,G) or ROS content reduction (Figure 5H,I). Overall, these results stress that oleuropein used at high concentrations acts as a pro-oxidant and, thus, as an anticancer compound in the MCF-7 breast cancer cell line; however, low doses seem to keep the redox state of MCF-7 cells uncharged, suggesting that oleuropein does not function as an antioxidant molecule in this cancer cell type.

## 3. Discussion

The Mediterranean diet has been related to lower morbidity and an overall improvement in health. Its benefits are often associated with the high content of polyphenols within olives and olive oil that are protective against several diseases, including those with cardiovascular, neurological, and metabolic origins [7,33,41]. 

Oxidative stress also plays a key role in the initiation and progression of carcinogenesis. Indeed, tumor cells show intracellular high levels of ROS due to an elevated metabolism rate or a metabolic reprogramming from oxidative phosphorylation (OXPHOS) [42], and this feature is often associated with key steps of carcinogenesis, including the induction of genetic alterations, cellular proliferation, the resistance to apoptosis, metastasis, and angiogenesis. For these reasons, the antioxidant activity of polyphenols has a potential role in chemoprevention. Indeed, polyphenols reduce DNA damage induced by carcinogens, act as ROS scavengers and metal chelators, or modulate the activity and expression level of oxidative stress-related enzymes [43,44]. Moreover, decreasing the intracellular ROS content by using antioxidant enzymes can reverse the malignant phenotype of cancer cells [45]. Intriguingly, it has been recently demonstrated that polyphenols may also have pro-oxidant properties by increasing ROS levels, and this has opened another unexpected opportunity to use these compounds as a potential therapeutic strategy against cancer. The use of dietary agents, such as curcumin and resveratrol, has been demonstrated to promote the production of hydrogen peroxide and to efficiently kill tumor cells without affecting normal cells [46].

It was shown that OLE concurrently exerts antioxidant and pro-oxidant activity.

However, further studies are necessary to better elucidate the role of OLE in the regulation of intracellular redox homeostasis in different contexts.

In this study, we demonstrated that in HEY ovarian cancer cells, the use of OLE over a concentration range of 100 μM to 400 μM, chosen in agreement with currently available studies in different cancer models such as pancreatic and leukemia [22,40,47], is associated with a reduction in cell growth and viability in a dose-dependent manner. In particular, our data indicate that at the highest concentration (400 μM), OLE determines an increase in ROS production and LIP levels thus breaking down the cell cycle S-phase and triggering apoptosis. In agreement with our results, it has been previously shown that in the OVCAR-3 ovarian cancer cell line, OLE induces cell cycle arrest through a decrease in Cyclin B2, leading to the G2 to M-cell cycle progression, in parallel with an increase in apoptosis-related markers [20]. 

Notably, our study shows that OLE plays a dual role in the regulation of LIP and ROS content. Indeed, when used at low concentrations (100 μM), OLE decreases both LIP and ROS contents in HEY cells. Then, to better dissect these apparently opposing effects, we increased intracellular iron levels of HEY cells by using erastin, a well-known ferroptosis inducer (FIN). Erastin is a small molecule that induces cancer cell death in an iron-dependent manner and its efficacy seems to be optimal in tumor cells harboring *KRAS* mutation, such as HEY cells [37,48]. 

Interestingly, our results revealed that when used at the lowest dose (100 μM), OLE decreases the accumulation of LIP and ROS content caused by erastin treatment and significantly counteracts cell death triggered by erastin-mediated oxidative stress. 

Here, for the first time, we demonstrated that low doses of OLE reduce mitoROS levels induced by erastin treatment and, in parallel, restore the protein expression levels of GPX4 otherwise repressed by erastin administration. Overall, these data suggest that, depending on the concentrations, OLE may play a dual role in oxidative stress in HEY cells. 

A question remains open: can OLE also have a double-edged sword role in other cancer cell types? Based on this question, we decided to test the effects of OLE treatment on a well-studied in vitro model of breast cancer, MCF-7 cells. It has been demonstrated that in MCF-7 cells, OLE functions as a pro-oxidant and that this promotes cell death [12]. Our data show that OLE inhibits cancer cell growth in a dose-dependent manner and induces cell death at high doses, in agreement with recent findings [39]. Indeed, high concentrations of OLE elicited cell cycle alteration and apoptosis, confirming previous findings on its pro-oxidant and anticancer activities [39,49]. Instead, unlike HEY ovarian cancer cells, no alterations in LIP and total ROS levels in breast cancer cells after OLE treatment were observed. 

In conclusion, the novelty of our study relies mainly on two aspects: (i) the effects of OLE on ovarian cancer have been poorly elucidated in previous findings and we now add significant results on the protective role of OLE against oxidative damage; and (ii) the antioxidant activity of OLE on the mitochondrial dysfunction induced by erastin was never assessed previously, supporting our finding on the ability of OLE to improve cell viability and proliferation in a cell-specific manner. 

In this view, we believe that these findings significantly expand our understanding of the molecular mechanism underlying OLE pro-survival activity, likely involving in vitro and in vivo studies.

## 4. Materials and Methods

### 4.1. Cell Lines and Cell Culture

The human epithelial ovarian cancer cell line HEY and the human breast cancer cell line MCF-7 were purchased from the American Type Culture Collection (ATCC, Rockville, MD, USA). HEY and MCF7 cells were cultured in DMEM supplemented with 10% fetal bovine serum, 50 U of penicillin, and 50 μg of streptomycin/mL (Thermo Fisher Scientific, Milan, Italy). The two cell lines were maintained at 37 °C in 5% CO_2_.

### 4.2. Reagents

Oleuropein was extracted from olive leaves of the Coratina cultivar of *Olea europaea* L. as reported by Procopio et al. [50]. Briefly, olive leaves were dried, milled, and extracted in an Anton Paar Synthos 3000 MW Oven, at 800 W (P-controlled mode) for 10 min, using water as solvent. Then, the leaves were filtered, the solution was dried under pressure, and acetone was added to the mixture. The solid residue was eliminated by filtration, the solution was evaporated under reduced pressure and the crude was purified by flash chromatography on silica cartridges (CH_2_Cl_2_/MeOH 8:2). Oleuropein was obtained at an HPLC purity of 98%. Analytical data of the pure oleuropein were compared with data reported in the literature. Erastin (E7781) was purchased from Sigma-Aldrich, St. Louis, MO, USA. For oleuropein and erastin treatments, HEY and MCF7 cells were plated at a density of 5.0 × 10^5^ cells/well in a 6-well plate in a complete medium. The next day, cells were treated with oleuropein (at 100, 200, and 400 µM for 24 h) and erastin at 2.5 µM for 24 h. The used OLE concentration and times of exposure agreed with previous literature studies [20,22,40,47].

### 4.3. Cell Viability, Cell Death, and Apoptosis Assay

The growth rates of HEY and MCF7 cells were obtained using the Trypan Blue dye exclusion method. Cells were counted after 24 h of treatment. The viability of HEY and MCF-7 cells was also evaluated using the Cell Titer-Glo Luminescent Cell Viability Assay (Promega, Milan, Italy) kit which allows the evaluation of cell proliferation based on their ATP consumption [51]. A total of 5.0 × 10^3^ cells/well were plated in 96 wells and then treated at the concentration and time indicated. Then, the substrate and the enzyme are added in a 1:1 ratio. The plate was then incubated at 37 °C for 30 min and the readings (GloMax Explorer, Promega, Madison, WI, USA) were taken after 24 h of growth. Cell death assay was performed as follows: HEY and MCF7 cell lines, 5.0 × 10^5^ cells were seeded in 6-well plates overnight and treated at concentrations and times indicated. Cells were incubated with PI staining in the dark at 37 °C for 15 min. Samples were then washed twice with PBS. A total of 2.0 × 10^4^ events were acquired by a FACS BD LSRFortessaTM X-20 cytofluorometer (BD Biosciences, Milan, Italy). PI-positive cells were analyzed by the FlowJo software program (Tree Star, Inc., Ashland, OR, USA). For the apoptosis assay, 5.0 × 10^5^ cells/well were seeded in 6-well plates overnight followed by the various treatments. Cells were centrifuged and the relative pellets were stained with FITC-conjugated Annexin V (Milteny Biotech, Bergisch Gladbach, Germany). A total of 2.0 × 10^4^ events were acquired by a FACS BD LSRFortessaTM X-20 cytofluorometer (BD Biosciences, Milan, Italy), and the FITC-positive fluorescence was analyzed by FlowJo software (Tree Star, Inc., Ashland, OR, USA) [52].

### 4.4. Cell Cycle Analysis

Cell cycle analysis was performed as previously described [53]. In brief, cells were fixed with 70% (*v/v*) cold ethanol and stored at −20 °C for 1 h. Then, cells were washed with cold PBS, centrifuged and the pellets were resuspended in 500 μL of a non-lysis solution containing 50 μg/mL PI and RNase 250 μg/mL. After incubation at 4° for 30 min, cells were analyzed with a FACS BD LSRFortessaTM X-20 cytofluorometer.

### 4.5. Measurement of Intracellular and Mitochondrial ROS 

Intracellular and mitochondrial ROS analyses were performed as previously described [37]. Briefly, the levels of intracellular ROS were determined by incubating cells for 10′ at 37 °C with the redox-sensitive probe 2′-7′-Dichlorodihydrofluorescein diacetate (CM-H2DCFDA; Thermo Fisher Scientific, Waltham, MA, USA) according to the instructions of the manufacturer. CM-H2DCFDA fluorescence was analyzed by flow cytometry using a FACS BD LSRFortessaTM X-20 cytofluorometer (BD Biosciences) and data were processed with FlowJo software (Tree Star, Inc.). After treatments, cells were incubated with 5 µM MitoSOX Red (MitoSOX Red Mitochondrial Superoxide Indicator, Thermo Fisher Scientific Inc.) for 10 min at 37 °C and then 2.0 × 10^4^ cells were acquired by flow cytometry using a FACS BD LSRFortessaTM X-20 cytofluorometer (BD Biosciences). Fluorescence data were processed with FlowJo software (Tree Star, Inc.).

### 4.6. Measurement of the Labile Iron Pool (LIP) Level

HEY and MCF7 cells were seeded in 6-well plates at a density of 5.0 × 10^5^ cells/well and grown overnight, and treated at the concentration indicated for 24 h. Cells were loaded with 0.25 μM calcein acetoxymethyl ester (CA-AM) (Sigma-Aldrich, St. Louis, MO, USA) for 30 min at 37 °C, then washed with PBS1X and treated or not with 200 mM 3-hydroxy-1,2-dimethyl-4(1H)-pyridone (deferiprone or L1) (Sigma-Aldrich, St. Louis, MO, USA). Following staining, and washing with PBS, cells were analyzed using a FACS BD LSRFortessaTM X-20 cytofluorometer (BD Biosciences). The ∆ mean fluorescence intensity (∆MFI) between chelator-treated and untreated cells reflected the amount of LIP [54].

### 4.7. Protein Extracts and Western Blotting 

Total extracts were prepared as previously described [55]. Briefly, to obtain total protein extracts, cells were washed once with PBS (1×) and total cell lysates were prepared using RIPA [56,57]. The samples were centrifuged at 12,000 rpm for 20 min at +4 °C and supernatants containing the total extracts were recovered. Proteins were separated on 4–12% NuPAGE Novex Bis-Tris protein gradient polyacrylamide gels (Thermo Fisher Scientific) and blotted onto nitrocellulose. Membranes were blocked with 5% milk (BioRad) and then incubated with the following antibodies: anti-GPX4 (1:500, ab41787), anti-GAPDH-HRP-conjugated (1:1000, sc-47724). Peroxidase AffiniPure Sheep Anti-Mouse IgG, Peroxidase AffiniPure Donkey Anti-Rabbit IgG, Peroxidase AffiniPure Donkey Anti-Goat IgG (1:10,000, Jackson ImmunoResearch Europe Ltd. Cambridge House) secondary antibodies were used. Signals were detected using the WESTARɳ2.0 ECL substrates for Western blotting (XLS070,0250, Cyanagen, Bologna, Italy) and acquired using a Uvitec Alliance Mini HD9 (Uvitec, Cambridge, UK).

### 4.8. Statistical Analysis 

Statistical analysis was performed as previously described [58]. Briefly, the analysis was performed using the two-tailed unpaired Student’s *t*-test using the GraphPad Prism^®^ software (San Diego, CA, USA) package. Statistical significance was determined by *p*-value < 0.05.

## Figures and Tables

**Figure 1 ijms-24-00842-f001:**
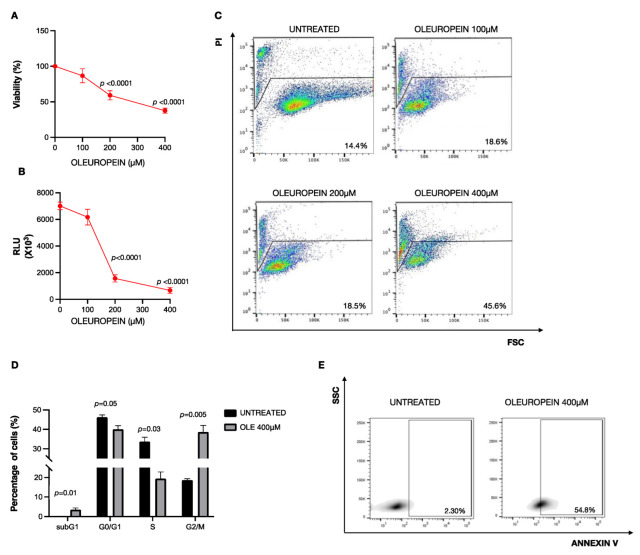
A high dose of oleuropein inhibits cell survival by increasing apoptosis. HEY cell viability and mortality were measured after 24 h of OLE treatment at the concentrations indicated in the figure by Trypan Blue dye exclusion (**A**), Cell Titer-Glo Luminescent cell viability assay (**B**), and by Propidium Iodide (PI) staining (**C**). (**D**) A bar diagram of the number of cells in each phase of the cell cycle based on PI staining and (**E**) representative density plot of Annexin V binding assay of in vitro-cultured untreated and treated HEY cells with OLE (400 μM) analyzed by flow cytometry. Values (mean ± SEM; *n* = 3) are shown. Statistically significant difference was determined by Student’s *t*-test.

**Figure 2 ijms-24-00842-f002:**
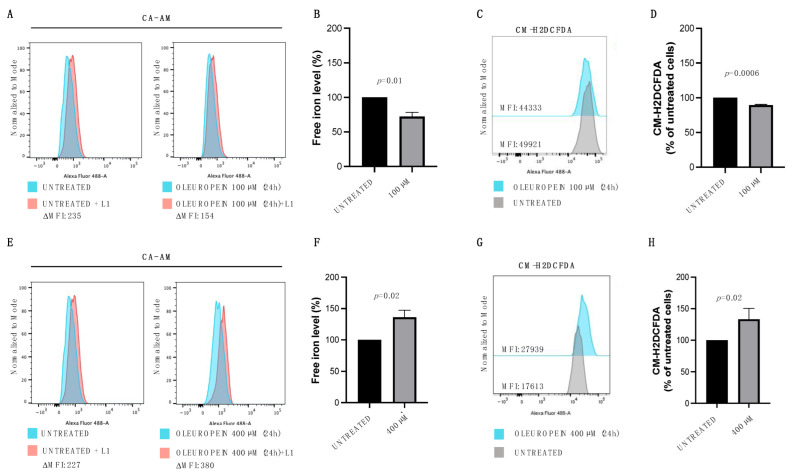
Pro- and antioxidant activity of oleuropein is dose-dependent. (**A**) Flow cytometry histogram showing LIP levels assessed using Calcein-AM assay in HEY cells untreated and treated with OLE (100 μM) for 24 h. (**B**) A bar diagram showing LIP levels presented as the % of control (untreated HEY cells). (**C**) Flow cytometry histogram showing the analysis of cytosolic ROS measured upon staining with CM-H2DCFDA in HEY cells untreated and treated with OLE (100 μM) for 24 h. (**D**) A bar diagram showing ROS levels presented as the % of control (untreated HEY cells). (**E**) Flow cytometry histogram showing LIP levels assessed using Calcein-AM assay in HEY cells untreated and treated with OLE (400 μM) for 24 h. (**F**) A bar diagram showing LIP levels presented as the % of control (untreated HEY cells). (**G**) Flow cytometry histogram showing analysis of cytosolic ROS measured upon staining with CM-H2DCFDA in HEY cells untreated and treated with OLE (400 μM) for 24 h. (**H**) A bar diagram showing ROS levels presented as the % of control (untreated HEY cells). Values (mean ± SEM; *n* = 3) are shown. Statistically significant difference was determined by Student’s *t*-test.

**Figure 3 ijms-24-00842-f003:**
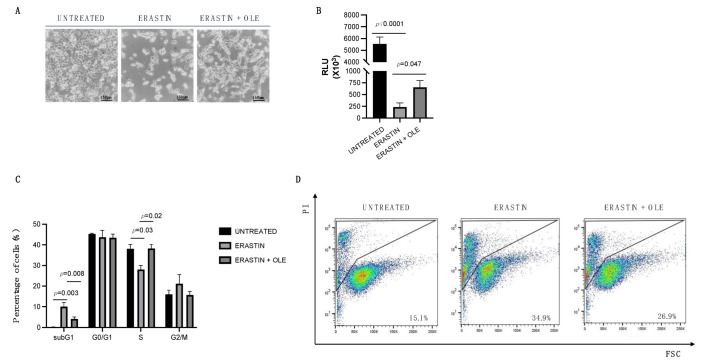
OLE administration protects HEY cells from cytotoxic effects induced by erastin. (**A**) Representative optical microscopy images of HEY cells untreated, treated with 2.5 µM erastin alone, or in combination with OLE (100 μM) for 24 h. Scale bars: 150 μm. Magnification: 100X. (**B**). A bar diagram of cell viability assessed by Cell Titer-Glo Luminescent cell viability assay of HEY cells untreated, treated with 2.5 µM erastin alone, or in combination with OLE (100 μM) for 24 h. (**C**) A bar diagram of the number of cells in each phase of the cell cycle based on PI staining. (**D**) Flow cytometric analysis showing PI staining of HEY cells untreated, treated with 2.5 µM erastin alone, or in combination with OLE (100 μM) for 24 h. Values (mean ± SEM; n = 3) are shown. Statistically significant difference was determined by Student’s *t*-test.

**Figure 4 ijms-24-00842-f004:**
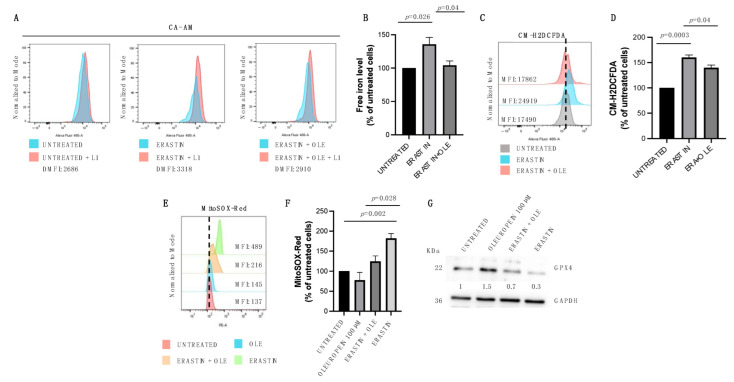
OLE administration mitigates intracellular iron and ROS accumulation induced by erastin. (**A**) Flow cytometry histogram showing LIP levels assessed using Calcein-AM assay in HEY cells untreated, treated with 2.5 µM erastin alone, or in combination with OLE (100 μM) for 24 h. (**B**) A bar diagram showing LIP levels presented as the % of control (untreated HEY cells). (**C**) Flow cytometry histogram showing the analysis of cytosolic ROS measured upon staining with CM-H2DCFDA in HEY cells untreated, treated with 2.5 µM erastin alone, or in combination with OLE (100 μM) for 24 h. (**D**) A bar diagram showing ROS levels presented as the % of control (untreated HEY cells). (**E**) Flow cytometry histogram showing the analysis of mitochondrial ROS (mitoROS) quantified by using MitoSOX Red of HEY cells untreated, treated with 2.5 µM erastin alone, or in combination with OLE (100 μM) for 24 h. (**F**) A bar diagram showing mitoROS levels presented as the % of control (untreated HEY cells). (**G**) Western blot analysis of GPX4 and GAPDH in HEY cells untreated, treated with 2.5 µM erastin alone, OLE (100 μM) alone, or in combination for 24 h. Densitometric values of GPX4 bands were normalized to GAPDH bands. Values (mean ± SEM; *n* = 3) are shown. Statistically significant difference was determined by Student’s *t*-test.

**Figure 5 ijms-24-00842-f005:**
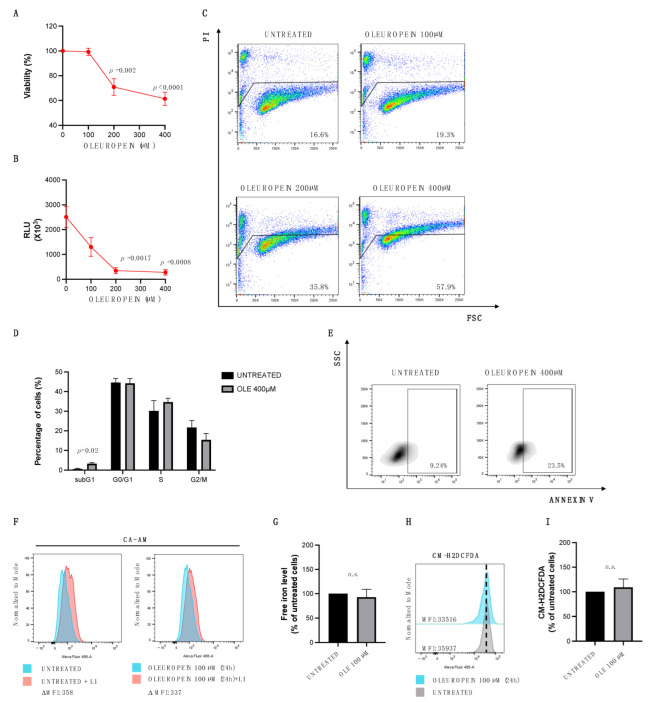
Oleuropein acts as a pro-oxidant in MCF-7 breast cancer cells. MCF7 cell viability and mortality were measured after 24 h of OLE treatment at the concentrations indicated in the figure by Trypan Blue dye exclusion (**A**), Cell Titer-Glo Luminescent cell viability assay (**B**), and by PI staining (**C**). (**D**) A bar diagram of the number of cells in each phase of the cell cycle based on PI staining and (**E**) representative density plot of Annexin V binding assay of in vitro-cultured untreated and treated MCF-7 cells with OLE (400 μM) analyzed by flow cytometry. (**F**) Flow cytometry histogram showing LIP levels assessed using Calcein-AM assay in HEY cells untreated and treated with OLE (100 μM) for 24 h. (**G**) A bar diagram showing LIP levels presented as the % of control (untreated HEY cells). (**H**) Flow cytometry histogram showing the analysis of cytosolic ROS measured upon staining with CM-H2DCFDA in MCF-7 cells untreated and treated with OLE (100 μM) for 24 h. (**I**) A bar diagram showing ROS levels presented as the % of control (untreated MCF-7 cells). Values (mean ± SEM; *n* = 3) are shown. Statistically significant difference was determined by Student’s *t*-test. Not statistically significant: *n.s.*

## Data Availability

The data presented in this study are available on request from the corresponding author.

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
