# Peer review of "The Double-Edged Sword of Oleuropein in Ovarian Cancer Cells: From Antioxidant Functions to Cytotoxic Effects"

_ijms, 2023, doi:10.3390/ijms24010842_

Round 1

Reviewer 1 Report

The paper is well written and is interesting for scholars of the topics as cancer, aging, and Mediterranean diet. In general, the referee has nothing to complain about the formulation of the problem, the methodologies used, the results obtained and the discussion of the results except for a point that the authors have not addressed, but which the referee deems necessary. A reader unfamiliar with the biological effects of phytochemicals could understand that double edged sword is a property of oleuropein alone, when in reality most of the phytochemicals of the Mediterranean Diet have different effects at low and high doses as clearly explained in the attached figure by  the referee with the relevant bibliographic entry. In the opinion of the referee, the authors should briefly discuss this aspect.

Author Response

Reviewer 1

Comments and suggestions for Authors:

The paper is well written and is interesting for scholars of the topics as cancer, aging, and Mediterranean diet. In general, the referee has nothing to complain about the formulation of the problem, the methodologies used, the results obtained and the discussion of the results except for a point that the authors have not addressed, but which the referee deems necessary. A reader unfamiliar with the biological effects of phytochemicals could understand that double edged sword is a property of oleuropein alone, when most of the phytochemicals of the Mediterranean Diet have different effects at low and high doses as clearly explained in the attached figure by the referee with the relevant bibliographic entry. In the opinion of the referee, the authors should briefly discuss this aspect.

Response 1: We thank the reviewer for his/her helpful suggestion. We added a brief sentence in the Introduction section as reported below:

Lines 63-67: The double-edged sword of OLE is a typical property of most phytochemicals of Mediterranean Diet. Evidence support that these natural compounds act as hormetins, triggering a biphasic dose-response of biological systems (Sawan Alì et al, Mechanisms of ageing and dev 200, 2021). It is demonstrated that low-dose exposures induce stimulatory or beneficial effects, while high-dose exposures lead to inhibitory or toxic effects (Mattson, M., Calabrese, E., 2009. Hormesis: What It Is and Why It Matters, pp. 1–13). 

Reviewer 2 Report

Title:  The double-edge sword of oleuropein in ovarian cancer cells: from antioxidant functions to cytotoxic effects

General comment

1- It is expected to be better interpreted in the discussion section according to the various parameters examined in this experiment. The discussion seems to be rewritten

2- The quality of the figures were low

 Specific Comments:

 Introduction

 Page 2 - Line 48: please add appropriate reference

 Page 2 - Line 59-60: Depend on this phrase, in this study author need to essay cell type, different time and OLE dosage

 Page 2 - Line 66: Please briefly clarify how?

  Result

 Page 5 - Line 173: figure is not clear (except figure 1, others (Figure 2-5) are not clear, please replace)

 Discussion

 Page 9 - Line 309: The discussion should be written based on the relevant mechanisms

Please use up-to-date reference

Page 9 - Line 322: Please use up-to-date reference

 Page 10 - Line 379: The discussion should be written based on the relevant mechanisms

 Page 10 - Line 389: Explain why negative and positive control were not used in this experiment

 Page 10 - Line 396:  how extracted? Please explain more...

 Page 10 - Line 401:  Explain clearly based on which procedure and references you have chosen different doses and exposure time of OLE for this experiment

References

Page 13 - Line 479: Need to recheck based on ijms format reference

Author Response

Reviewer 2

General comment

1- It is expected to be better interpreted in the discussion section according to the various parameters examined in this experiment. The discussion seems to be rewritten

2- The quality of the figures was low

 Specific Comments:

Introduction

Page 2 - Line 48: please add appropriate reference

Response 1: We thank the reviewer for his/her helpful suggestion. An appropriate reference has been added: “Neuroprotective effects of oleuropein; Recent developments and contemporary” M. S. Butt, U. Tariq, I.-Ul-Haq, A. Naz, M. Rizwan J Food Biochem. 2021, 45:e13967. DOI: 10.1111/jfbc.13967.

Page 2 - Line 59-60: Depend on this phrase, in this study author need to essay cell type, different time and OLE dosage

Response 2: We thank the reviewer for his/her observation. Indeed, currently, in this study we assessed the effect of OLE on two different cell lines representative not only of two different tumor types but also of two different tissues of origin. The effects on cell viability were tested at different concentrations chosen according to preliminary data already published by other research groups and used to cover a wide range of dosages. Based on these results we then decided to assess the pro- or anti-oxidant function of OLE. Demonstrating that OLE has a typical behavior of hormetins, as suggested by the first reviewer and explained in the new lines 63-67 of the revised manuscript. Of course, we totally agree with the reviewer that other cell types and treatments are mandatory to expand our study and to confirm the hypothesis arising from this pilot study.

Page 2 - Line 66: Please briefly clarify how?

Response 3

According to Alirezaei M. et al (J Physiol Biochem, 2012), the OLE-induced activation of GPX appears mediated by its capacity to increase the intracellular amount of the reduced glutathione (GSH), which acts as major co-factor of this antioxidant enzyme. The effects of OLE on CAT and SOD, instead, remain to be defined.

See new lines 74-77. 

Result

Page 5 - Line 173: figure is not clear (except figure 1, others (Figure 2-5) are not clear, please replace)

Response 4: Thank the reviewer for her/his helpful suggestion. We have now improved the overall quality of the figures.

Discussion

Page 9 - Line 309: The discussion should be written based on the relevant mechanisms. Please use up-to-date reference.

Response 5: We thank the reviewer for his/her valuable comment. We modified the main text at the Discussion section as reported at the pages 9-10 of the revised manuscript.

Line 276: Reference has been replaced by new references:

  1. Aiello, G.D. Guccione, G. Accardi, C. Caruso What olive oil for healthy ageing? Ma-turitas, 2014;
  2. Yoshioka, Y., et al., Anti-Cancer Effects of Dietary Polyphenols via ROS-Mediated Pathway with Their Modulation of MicroRNAs. Molecules, 2022. 27(12).

Page 9 - Line 322: Please use up-to-date reference

Response 6: Line 276: Reference has been replaced by new references:

  1. Aiello, G.D. Guccione, G. Accardi, C. Caruso What olive oil for healthy ageing? Maturitas, 2014.
  2. Yoshioka, Y., et al., Anti-Cancer Effects of Dietary Polyphenols via ROS-Mediated Pathway with Their Modulation of MicroRNAs. Molecules, 2022. 27(12).

Page 10 - Line 379: The discussion should be written based on the relevant mechanisms

Response 7. We thank the reviewer for his/her valuable comment. We modified the main text at the Discussion section as reported in Response 5.

Page 10 - Line 389: Explain why negative and positive control were not used in this experiment

Response 8: In this study, the authors consider the untreated cells as negative control of the OLE treatment. Once it is established that OLE acts as antioxidant and iron chelator at low concentrations, we proceeded to confirm its function by treating HEY cells with erastin (positive control of oxidative damage) in order to induce intracellular iron overload and, consequently, ROS production.We choose this molecule as positive control because its efficacy in inducing oxidative damage has been largely proved to be optimal in tumor cells harboring KRAS mutation such as HEY cells (Zhao, Y.C.; Li, Y.Q.; Zhang, R.F.; Wang, F.; Wang, T.J.; Jiao, Y. The Role of Erastin in Ferroptosis and Its Prospects in Cancer Therapy. Oncotargets Ther 2020, 13, 5429-5441, doi:10.2147/Ott.S254995. Battaglia, A.M.; Sacco, A.; Perrotta, I.D.; Faniello, M.C.; Scalise, M.; Torella, D.; Levi, S.; Costanzo, F.; Biamonte, F. Iron Administration Overcomes Resistance to Erastin-Mediated Ferroptosis in Ovarian Cancer Cells. Front Oncol 2022, 12, 868351, doi:10.3389/fonc.2022.868351.)

Page 10 - Line 396: how extracted? Please explain more...

Response 9: We thank the reviewer for his/her suggestion.  We added a brief sentence in the Materials and Methods section as reported below:

Line 339-345: Briefly, olive leaves were dried, milled and extracted in a Anton Paar Synthos 3000 MW Oven, at 800W (P-controlled mode) for 10 minutes, using water as solvent. Then, leaves were filtered, the solution was dried under pressure, and acetone was added to the mixture. The solid residue was eliminated by filtration, the solution was evaporated under reduced pressure and the crude was purified by flash chromatography on silica cartridges (CH2Cl2/MeOH 8:2). Oleuropein was obtained at HPLC purity of 98%. 

Page 10 - Line 401: Explain clearly based on which procedure and references you have chosen different doses and exposure time of OLE for this experiment

Response 10: We thank the reviewer for his/her suggestion. The used OLE concentration and times of exposure that we have chosen, agree with the literature studies on different cancer models such as breast cancer and leukemia. The references are reported below:

  1. H. Elamin, M.H. Daghestani, S.A. Omer, M.A. Elobeid, P. Virk, E.M. Al-Olayan, Z.K. Hassan, O.B. Mohammed, A. Aboussekhra, Olive oil oleuropein has anti-breast cancer properties with higher ef ciency on ER-negative cells, Food Chem. Toxicol. 53 (2013);
  2. Samet, J. Han, L. Jlaiel, S. Sayadi, H. Isoda, Olive (Olea europaea) leaf extract induces apoptosis and Monocyte/Macrophage differentiation in human chronic myelogenous leukemia K562 cells: insight into the underlying mechanism, Oxid. Med. Cell. Longev. 2014 (2014) 1–16;
  3. Goldsmith, Q. Vuong, E. Sadeqzadeh, C. Stathopoulos, P. Roach, C. Scarlett, Phytochemical properties and anti-proliferative activity of Olea europaea L. Leaf extracts against pancreatic cancer cells, Molecules 20 (2015) 12992–13004;
  4. Benot-Dominguez, R., et al., Olive leaf extract impairs mitochondria by pro-oxidant activity in MDA-MB-231 and OVCAR-3 cancer cells. Biomed Pharmacother, 2021. 134: p. 111139.

We added a brief sentence in the Materials and Methods section as reported below:

Line 350: The used OLE concentration and times of exposure agree with previous literature studies (ref).

References

Page 13 - Line 479: Need to recheck based on ijms format reference

Response 11: Done.